# Dual-Energy Computed Tomography in Detecting and Predicting Lymph Node Metastasis in Malignant Tumor Patients: A Comprehensive Review

**DOI:** 10.3390/diagnostics14040377

**Published:** 2024-02-09

**Authors:** Mengting Chen, Yundan Jiang, Xuhui Zhou, Di Wu, Qiuxia Xie

**Affiliations:** Department of Radiology, The Eighth Affiliated Hospital, Sun Yat-sen University, Shenzhen 518036, China; chenmt35@mail2.sysu.edu.cn (M.C.); jiangyd7@mail.sysu.edu.cn (Y.J.); zhouxuh@mail.sysu.edu.cn (X.Z.)

**Keywords:** DECT, metastatic lymph nodes, cancer, radiomics, artificial intelligence, deep learning

## Abstract

The accurate and timely assessment of lymph node involvement is paramount in the management of patients with malignant tumors, owing to its direct correlation with cancer staging, therapeutic strategy formulation, and prognostication. Dual-energy computed tomography (DECT), as a burgeoning imaging modality, has shown promising results in the diagnosis and prediction of preoperative metastatic lymph nodes in recent years. This article aims to explore the application of DECT in identifying metastatic lymph nodes (LNs) across various cancer types, including but not limited to thyroid carcinoma (focusing on papillary thyroid carcinoma), lung cancer, and colorectal cancer. Through this narrative review, we aim to elucidate the clinical relevance and utility of DECT in the detection and predictive assessment of lymph node metastasis in malignant tumors, thereby contributing to the broader academic discourse in oncologic radiology and diagnostic precision.

## 1. Introduction

The systematic evaluation of lymph nodes (LNs) is integral to the staging process in oncology, significantly impacting the determination of the therapeutic approaches. Lymph node metastasis often represents the initial stage of cancer metastasis, with cancer cells frequently disseminating to other organs via the lymphatic system. The presence and extent of lymph node metastasis are closely associated with patient prognosis and survival outcomes [1]. Therefore, it is crucial to precisely assess the state of the lymph nodes for accurate cancer staging and informed treatment planning.

Conventional computed tomography (CT) is the first-line modality for assessing primary tumor lesions and LN metastasis. This imaging method primarily depends on the morphological characteristics of LNs to distinguish metastatic from non-metastatic LNs. Such characteristics include LN size, degree of enhancement, and presence of necrosis or cystic degeneration [2,3,4]. However, instances of small LNs or those without typical morphological features have been retrospectively diagnosed as metastatic upon pathological examination [5], thereby highlighting the inherent limitations in the diagnostic efficacy of conventional CT. 

Dual-energy computed tomography (DECT), an advanced imaging technology, offers potential improvements in diagnosing LN metastasis. DECT uses two different X-ray energy levels, enabling enhanced material differentiation and quantification [6]. In recent years, DECT has been widely used in various types of cancers, including thyroid carcinoma, lung cancer, colorectal cancer, etc., showing promise in identifying LN metastasis across various malignances. Additionally, a few studies have explored combining DECT with artificial intelligence (AI) or radiomics features to predict LN metastasis yielding promising results. 

This review aims to provide an overview of DECT, focusing on its application in detecting metastatic LNs across various cancer types, and highlighting the potential of integrating DECT with advanced technologies like AI in oncological imaging.

## 2. Physics and Technology of DECT

DECT operates on the principle that imaging data acquired with high (120–150 kV) and low energy (80–100 kV) can effectively decompose and characterize materials [7]. In contrast to conventional CT, which uses a single X-ray energy spectrum, DECT captures two distinct datasets, enabling the decomposition and characterization of materials. This dual-energy approach facilitates enhanced tissue differentiation and provides incremental diagnostic information. Currently, there are six primary technical approaches to obtain DECT data: dual source, rapid voltage switching with a single tube, dual-layer detector with a single tube, single tube with a split filter, single tube with sequential dual scans, and photon-counting CT (PCCT) [8,9]. Additionally, several advanced post-processing techniques are integral to DECT, including virtual monoenergetic imaging (VMI), material decomposition (MD) images such as iodine maps, effective atomic number maps, electron density maps, and virtual non-contrast (VNC) [6]. DECT provides various quantitative parameters for material quantification, which will be extensively discussed in the next section. Relevant quantitative and qualitative parameters are summarized in Table 1. Post-processing techniques of DECT are summarized in Table 2.

## 3. Applications of DECT

### 3.1. Iodine Maps and Iodine Quantification

The utilization of iodine contrast in contrast-enhanced CT is paramount for assessing blood perfusion. The perfusion pattern is an important diagnostic marker in oncology [15]. Iodine maps, generated from contrast-enhanced DECT images through material decomposition post-processing techniques, display the attenuation characteristics attributable to iodine. Iodine maps serve not only as quantitative indicators of blood supply to tissues, but also provide insights into the angiogenesis and hemodynamic status of lesions, which are crucial for the assessment of tumor proliferation and metastatic potential. 

LNs are small bean-shaped organs that lie along lymph vessels and consist of the cortex and the medulla. The cortex contains collections of lymphocytes, predominantly B lymphocytes in follicular spaces and T lymphocytes in parafollicular spaces. The medulla provides a pathway for the afferent and efferent lymphatic and blood vessels [19]. Metastatic tumor cells often first appear in the LNs’ marginal sinus, from which they reach the medullary sinus, the medulla and the cortex, eventually leading to total parenchymal replacement [20]. This seeding and growth of tumor cells within LNs induce neoangiogenesis, possibly resulting in higher iodine concentrations in metastatic nodes. However, metastatic LNs often have necrosis and liquefaction, which may lead to lower iodine concentrations. When metastatic tumor cells replace the normal structure, the hilar structures can be compressed to the periphery of LNs [21].

Iodine concentration (IC) and normalized iodine concentration (NIC) are parameters derived from iodine maps, which have been wildly used to detect LN metastasis in patients with malignant tumors (Figure 1A,B). Retrospective studies by Kato et al. and Liu et al. [10,22] have shown that IC and NIC are significantly lower in metastatic LNs than in non-metastatic ones, with IC in the portal venous phase (PP) being a particularly strong indicator. After excluding LNs with clear metastatic features identifiable through CT, Kato et al. [10] found that 5 of the remaining 50 LNs were indeed metastatic, with IC in PP remaining the most robust predictor for metastatic LNs (cutoff: 2.1 mg/mL, area under the curve = 0.933). Liu et al. [22] enhanced detection by combining NIC in PP with the short-axis diameter of LNs, which increased the overall accuracy to 82.9%. Furthermore, a meta-analysis by Kong et al. [15] evaluated the diagnostic accuracy of contrast-enhanced DECT for detecting metastatic LNs in patients with cancer. This study, which reviewed 16 studies encompassing 984 patients and 2577 LNs, revealed that NIC in the arterial phase, when used in conjunction with the arterial phase slope, significantly improved the identification of metastatic LNs, achieving a sensitivity of 94%, specificity of 74%, and an area under the curve (AUC) of 0.94.

The majority of CT studies focus on LNs with a maximal short diameter greater than 5 mm, primarily due to the challenges of measuring smaller LNs resulting from the limited spatial resolution of CT. However, it has been reported that some small LNs were ultimately identified as metastatic LNs upon pathological examination [5]. Several studies have thus aimed to investigate the diagnostic efficacy of DECT for micro-metastatic LNs, defined as having a maximal short diameter within the range of 2–6 mm [23,24]. Micro-metastatic LNs or small LN metastasis can be considered as an early stage of LN metastasis. Zou et al. [23] conducted quantitative evaluations on LNs smaller than 5 mm in patients with papillary thyroid carcinoma (PTC) and demonstrated that DECT exhibited a good diagnostic performance in detecting metastatic LNs. The optimal parameter for diagnosing LN metastasis was IC in the arterial phase, with an AUC of 0.775. When the diameter, IC in the arterial phase, and NIC in the venous phase were combined, the diagnostic accuracy improved significantly, with an AUC reaching 0.819. Zhuo et al. [24] further corroborated these findings, showing that DECT, particularly at a small field of view (FOV), can accurately detect small LN metastasis in PTC.

Recent studies have also focused on novel parameters derived from iodine maps to evaluate their efficacy in diagnosing cervical LN metastasis from PTC [25,26]. Zhou et al. [25,26] introduced two such parameters: the DECT-derived extracellular volume (ECV) fraction and the arterial enhancement fraction (AEF) value. The arterial enhancement fraction (AEF) was defined as the ratio of iodine uptake in the arterial phase (AP) to that in the venous phase (VP), multiplied by 100% [11]. Their research concluded that both the ECV fraction and AEF value were significantly elevated in metastatic LNs compared with non-metastatic ones in patients with PTC. These initial findings suggest a potential for these quantitative parameters to enhance diagnostic accuracy, particularly when combined with other imaging modalities, warranting further investigation to substantiate their clinical utility.

### 3.2. Virtual Monochromatic Imaging (VMI)

VMI is monochromatic imaging that is virtually synthesized from dual-energy data and is also referred to as “monoenergetic imaging”. VMI can be generated within the 40–190 keV range and is renowned for its ability to optimize image noise and contrast while allowing precise monoenergetic contrast attenuation measurements [17]. 

In a pivotal study, Hu et al. [27] evaluated 74 mediastinal LNs in lung cancer patients, which included 33 metastatic and 41 non-metastatic LNs. This study focused on collecting attenuation values at the lower energy levels of VMI (40–90 keV). The study identified the attenuation value at 40 keV as the most effective biomarker for the diagnosing of mediastinal LN metastasis in non-small-cell lung cancer (NSCLC) with an AUC of 0.91, demonstrating high sensitivity (0.94) and specificity (0.81). Similarly, Sekiguchi et al. [28] confirmed the significant diagnostic value of VMI at 40 keV, particularly for evaluating lung hilar LNs.

Beyond individual diagnostic parameters, VMI has also been integral in the development of clinical–radiomics nomograms aimed at predicting LN metastasis. Lu et al. [29] developed a clinical–radiomics nomogram to predict cervical LN metastasis in patients with PTC. This model utilized radiomics features derived from DECT images at 80 kV of the entire thyroid tissue. It is noteworthy that CT images at this voltage offer higher contrast and a lower signal-to-noise ratio, making vein-phase CT images more representative of blood flow in the thyroid gland and primary lesions. 

### 3.3. Spectral Hounsfield Unit Attenuation Curves

Spectral Hounsfield unit attenuation curves serve as a quantitative measure correlating with different energy levels in VMI. These curves represent the energy-dependent changes in attenuation within a region of interest, typically spanning from 40 to 140 keV [18]. Notably, the spectral Hounsfield unit curve varies across different tissues. The slope of the spectral Hounsfield unit curve (λHu) is particularly valuable for component analysis and differential diagnosis (Figure 1G,H). Recent studies have focused on λHu for detecting LN metastasis in patients with malignant tumors. A meta-analysis by Wang et al. [30] evaluated the utility of quantitative spectral CT parameters in identifying LN metastasis in lung cancer. Incorporating 11 studies with 1290 cases, the analysis revealed that the diagnostic performance of NIC and λHu in identifying lymphatic metastasis surpassed that of short-axis diameter, with a combined AUC exceeding 0.8. Yang et al. further demonstrated that DECT quantitative parameters offered greater accuracy than conventional CT morphological assessments [10]. Their retrospective study included 84 patients with lung cancer, evaluating a total of 144 LNs, of which 48 were metastatic. The study highlighted that when λHU was set at an optimal threshold of 2.75, the AUC in the diagnosis of metastatic LNs was 0.951, notably higher than the AUC of 0.780 achieved by conventional CT assessments based on size. Moreover, a prospective study by Zhang et al. [31] investigated the diagnostic performance of quantitative parameters derived from DECT for preoperatively identifying metastatic sentinel lymph nodes (SLNs) in patients with breast cancer. The study involved patients undergoing dual-phase contrast-agent-enhanced CT. The findings underscored that venous phase λHu was the most effective single parameter for the detection of metastatic SLNs, achieving an AUC of 0.88. 

### 3.4. Effective Atomic Number Z_eff_

When dealing with radiation–matter interaction processes, the definition of Z_eff_ for compounds or mixtures (and with heterogeneous materials in general) involves the creation of a fictitious element with atomic number Z_eff_; the interaction cross sections for photoelectric effect and Compton scattering can be approximately expressed as proportional to Z_eff_^n^, where n is between 4 and 5 for photoelectric effect and 1 for Compton [13]. Z_eff_ can be calculated from a Z_eff_ map, which is a quantitative approach used for tissue characterization (Figure 1C,D). In a prospective single-center study by Yang et al. [16], 178 of the largest LNs (72 metastatic, 106 non-metastatic) identified from 178 patients with colon or high rectal cancer were included. Each patient underwent triphasic contrast-enhanced DECT. The study revealed that the most efficient DECT parameter to distinguish between metastatic and non-metastatic LNs was the normalized Z_eff_ (Z_eff-LN_/Z_eff-aorta_) during the portal venous phase. The parameter achieved an AUC of 0.871 and an accuracy of 84.8%, outperforming the traditional morphological features and short-axis diameter used in conventional CT.

### 3.5. Electron Density (ED) 

Electron density (ED) is the average number of electrons in a volume unit (typically expressed in e/cm^3^). ED especially plays an important role in radiation therapy treatment planning, where the accurate estimations of CT-derived ED maps are used by TPS software for calculating dose distributions. Even though ED can be one-to-one mapped to HU in single energy CT, DECT allows better quantification of ED [12]. Luo et al. [32] demonstrated that in a model incorporating arterial phase CT attenuation on 70-keV images, ED (VP) and clustered features achieved a higher AUC of 0.907 for LN diagnosis in gastric cancer, significantly outperforming conventional CT criteria. In 2021, Qiu and his team expanded the analysis to include more quantitative parameters including λHu, NIC, iodine water ratio (nIWR), ED, and Z_eff_ in patients with colorectal cancer [33]. Their study revealed that metastatic nodes exhibited significantly higher λHu, NIC, and nIWR values than non-metastatic nodes in both arterial and venous phases. However, ED and Z_eff_ did not show significant difference, a variation potentially attributable to the primary tumor’s nature, the DECT scanner used, or the nodal size. 

### 3.6. Multi-Parameter Evaluation

The superiority of DECT over conventional CT is largely attributed to its ability to provide a broader array of quantitative parameters for tissue characterization. Employing a multi-parameter evaluation approach significantly enhances the diagnostic efficacy for identifying metastatic LNs in patients with malignant tumors.

In a prospective study, Liu et al. [34] analyzed 45 patients with PTC, encompassing 63 metastatic and 112 non-metastatic LNs. The study examined the relationship between LN metastasis, simple DECT parameters, and qualitative CT features. They underwent DECT dual-phase post-contrast scans. The researchers found that a combination of venous phase λHu and arterial phase NIC values yielded substantially higher accuracy for the preoperative diagnosis of cervical nodal metastasis in patients with PTC than conventional CT imaging features such as nodal size and enhancement degree. Wu et al. performed a retrospective study aimed to assess the efficacy of integrating quantitative DECT parameters with qualitative morphological parameters for the preoperative prediction of cervical nodal metastasis in patients with PTC [5]. A total of 80 metastatic and 126 benign LNs from 35 patients were included. They concluded that the combination of quantitative DECT parameters (IC, Z_eff_, λHU, NIC) and morphological data (shortest diameter and pronounced enhancement) significantly improved the diagnostic performance compared with the use of any single parameter independently, achieving an AUC of 0.878. The corresponding sensitivity, specificity, accuracy, positive predictive value (PPV), and negative predictive value (NPV) of this combination were 86.3%, 72.2%, 77.7%, 66.3%, and 89.2%, respectively. Additionally, Zeng et al. [35] evaluated the diagnostic potential of DECT in the regional LN assessment for liver cancer patients. This study indicated that the combination of IC, NIC, and λHu values in the PP was superior to using any single parameter alone. Importantly, the presence of active hepatitis did not impede the DECT’s capability to characterize metastatic LNs effectively.

### 3.7. Comparative Efficacy of DECT and Other Radiological Modalities

In the contemporary landscape of noninvasive radiological diagnostics, modalities such as endoluminal ultrasound (US), CT, and magnetic resonance imaging (MRI) have been widely used in the assessment of LN metastasis. Particularly in the context of rectal carcinoma, MRI is often the recommended choice [36,37]. However, DECT, with its advanced capabilities for spectral evaluation and material-specific tissue characterization, has demonstrated outstanding diagnostic efficacy in detecting metastatic LNs compared with US, CT, and MRI. 

In the realm of thyroid disease, US remains the preferred preoperative imaging modality. However, its operator dependency and limited capacity to visualize deep-seated LNs may compromise the accurate diagnosis of cervical nodal metastasis in PTC patients [2,38]. Li et al. [2] compared the diagnostic performance of DECT and US in detecting lateral cervical nodal metastasis in PTC. The study revealed that combined DECT parameters (AUC = 0.942) significantly surpassed US morphological parameters (AUC = 0.771, *p* < 0.001) in diagnostic accuracy, with a sensitivity, specificity, and accuracy of 92.9%, 86.2%, and 90.9%, respectively. Similarly, the retrospective assessment by Yoon et al. [38] of 102 patients (49 with LN metastases and 53 without) indicated that integrating DECT parameters with US features elevated the diagnostic AUC from 0.890 to 0.941. 

Contrastingly, a comparative study between DECT and 18F-FDG PET/CT in primary tumors and LNs of lung cancer showed no correlation between the two in primary tumors and metastatic LNs [39]. This suggests that DECT and 18F-FDG PET/CT elucidate different tumor and nodal characteristics and should be considered complementary rather than substitutive. A retrospective study by Nagano et al. [40] aimed to assess the utility of ED from DECT in diagnosing metastatic mediastinal LNs in patients with NSCLC, in comparison with conventional CT and FDG PET/CT. This study found that the ED of metastatic nodes was significantly lower than that of non-metastatic nodes, and combinations of ED with short-axis diameter or positive FDG uptake outstripped individual parameters in accuracy, reaching 82.9% and 82.1%, respectively. 

Ai et al. [41,42] investigated the accuracy of LN staging in rectal cancer using isolated surgical specimens and discovered no significant differences between DECT and MRI. The DECT evaluation reduces the problem of individual judgement and a high observer variation, as seen in MRI. Additionally, DECT may improve staging in the less highly dedicated centers. Thus, the combination of MRI and DECT may provide additional insights in cases of Nx staged patients. Direct application of X-rays to specimens offered more immediate access to LNs, yielding a more accurate DECT depiction. Nonetheless, comparative in vivo studies between DECT with MRI in rectal cancer are relatively scarce, indicating a need for further research.

PCCT is a novel technical approach to obtain DECT data, which uses special X-ray detectors capable of collecting incoming X-ray photons in ≥2) energy bins, without any need for kVp switching of split filters. Yalon et al. [43] investigated the feasibility and performance of PCCT for detecting breast cancer and nodal metastases and found that PCCT showed initial promising results in characterizing breast cancer and regional lymphadenopathy similar to MRI. However, this study was limited by the small number of subjects (13 patients) and further research is needed in this domain.

## 4. Limitation

While DECT shows considerable promise as a preoperative evaluation tool for metastatic LNs in patients with malignant tumors, it is still in the early stages of adoption and research. Several limitations warrant attention and further investigation. 

Primarily, the majority of existing studies are conducted within single centers and are retrospective in nature. This, coupled with the variation in DECT equipment across different institutions, may compromise the generalizability and reliability of the results. Consequently, there is a compelling need for future multi-center, large cohort studies to substantiate the findings and enhance the credibility of DECT as a diagnostic tool. 

Moreover, comprehensive prospective studies are essential to compare DECT with other imaging modalities such as PET/CT, MRI, and photon-counting CT. Such research would provide valuable insights into the clinical application and broader utilization of DECT. 

Inconsistencies in the literature regarding DECT parameters, such as Z_eff_ and ED, also pose significant challenges. While some studies note a marked difference between metastatic and non-metastatic LNs [34,40], others, like Qiu et al.’s study, presented contrary conclusions [33]. These discrepancies may be attributed to variations in DECT scanners, nodal sizes, and the characteristics of the primary tumor. This highlights the need for further research with expanded sample sizes to elucidate whether DECT parameters exhibit varied sensitivity or specificity in detecting metastatic LNs across different malignancies. 

Additionally, conflicting findings regarding iodine concentrations in metastatic LNs, as evidenced in Chen’s study compared to others [44], raise questions about the consistency of LNs across studies. The frequent occurrence of necrosis and liquefaction in metastatic LNs could contribute to these inconsistencies. Deep learning and AI models based on DECT radiomics may offer potential solutions to these challenges.

## 5. Radiomics and Artificial Intelligence in DECT

Radiomics, aiming to extract clinical information through quantitative data (features) from medical images [45,46], has been revolutionized by Artificial Intelligence (AI). AI allows for the extraction of hundreds of radiomics features from a region or volume of interest (ROI/VOI) which are then analyzed using high-order statistical methods with machine learning (ML) and deep learning (DL) to correlate with clinical outcomes [47,48,49]. 

Previous studies have indicated that DECT-derived quantitative parameters are promising in discriminating metastatic cervical LNs based on average iodine uptake. However, these studies often overlook the heterogeneity of LNs related to necrosis, extracellular mucin, or calcifications, which are crucial in identifying metastatic LNs [50]. Radiomics analysis can address this shortfall by enabling noninvasive profiling of tumor heterogeneity [45,51]. A retrospective study by Zhou et al. [50] applied radiomics analysis based on iodine maps to analyze cervical metastatic LNs from PTC, revealing that radiomics analysis of DECT-derived iodine maps outperformed qualitative CT image evaluations, especially when combined with CT image features. 

Studies have also extended radiomics analysis to primary lesions and entire thyroid tissues to predict LN metastasis. Zhou et al. [52] developed and validated radiomics nomograms based on iodine maps for preoperative prediction of cervical and central LN metastasis in PTC, demonstrating good discrimination and calibration in both the training (AUC = 0.847, 0.837) and the validation cohorts (AUC = 0.807, 0.795), especially in CT reported LN negative groups. Significant improved AUC, net reclassification index (NRI), and integrated discriminatory improvement (IDI) substantiated the enhanced predictive value of the two rad scores when compared with clinical models without radiomics. 

Comparatively, radiomics parameters provide more quantitative information than traditional parameters used by radiologists. In 2022, Wang et al. [53] explored the relationship between DECT radiomics features of regional largest short-axis LNs and metastasis in patients with rectal cancer. After post-processing the venous phase images from DECT, 120 kVp-like images and iodine images were obtained, from which a substantial set of 833 features were extracted. The radiomics features derived from both the 120 kVp-like images and iodine maps showed excellent diagnostic performance in the test group, achieving AUC values of 0.922 and 0.866, respectively. Notably, when compared with conventional DECT quantitative parameters and iodine maps, the predictive radiomics model based on 120 kVp images in conjunction with the largest short-axis diameter of the LN emerged as the most valuable tool for predicting LN metastasis in patients with rectal cancer.

The development of AI has also let to the construction of deep learning models to accurately predict metastatic LNs in lung cancer, with one team achieving up to 93% accuracy [54,55]. These models utilized lower energy-level images to train the fusion model, which provided high contrast suitable for discriminating LN metastasis. This suggests that tumor heterogeneity and size are significant factors in the model’s ability to determine the presence or absence of nodal metastasis from the primary tumor. 

Further, recent studies have integrated DECT images with deep learning, AI, and radiomics features to construct sophisticated prediction models [56,57,58,59]. In 2022, An et al. [56] built a deep learning radiomics model using DECT, achieving an AUC of 0.92 in predicting LN metastasis in pancreatic ductal adenocarcinoma. The models utilized two sets of DECT images (100 and 150 keV) along with selected clinical variables. Such prediction models are instrumental in clinically detecting LN metastasis and stratifying patients at risk, thereby guiding treatment planning. In 2023, Bian et al. [60] developed and validated an automated preoperative AI algorithm for tumor and LN segmentation using CT imaging to predict LN metastasis in patients with pancreatic ductal adenocarcinoma. Their findings indicated that the AI model outperformed radiologists, as well as clinical and radiomics models, in predicting LN metastasis using conventional CT. However, AI models based on DECT are still in their nascent stages, and there is a clear call for more multi-center studies. 

Theoretically, the combination of DECT, deep learning methodologies, and predictive models has the potential to improve the preoperative predictive performance for LN metastasis across various types of tumors. 

## 6. Conclusions

The N stage is critical in the management of malignant tumors, and extensive research has explored the capabilities of DECT for detecting and predicting LN metastasis in patients with such conditions. In contrast to conventional CT, DECT offers not only a range of quantitative parameters, but also superior quality images across various voltage levels, showcasing its considerable potential as an innovative technology. However, previous studies exhibit a degree of heterogeneity due to the diversity of cancer types studied and the different models of equipment employed. There is a pressing need for further research to establish standardized DECT quantitative parameters for LN metastasis, which may expand its clinical utility in the future. At the same time, the importance of dose optimization should be constantly considered. Notably, recent advancements in radiomics and AI have opened up new avenues. Previous studies have highlighted the benefits of integrating these cutting-edge approaches with DECT. Moving forward, this promising direction warrants continued exploration to further optimize the diagnostic performance of DECT in oncological settings.

## Figures and Tables

**Figure 1 diagnostics-14-00377-f001:**
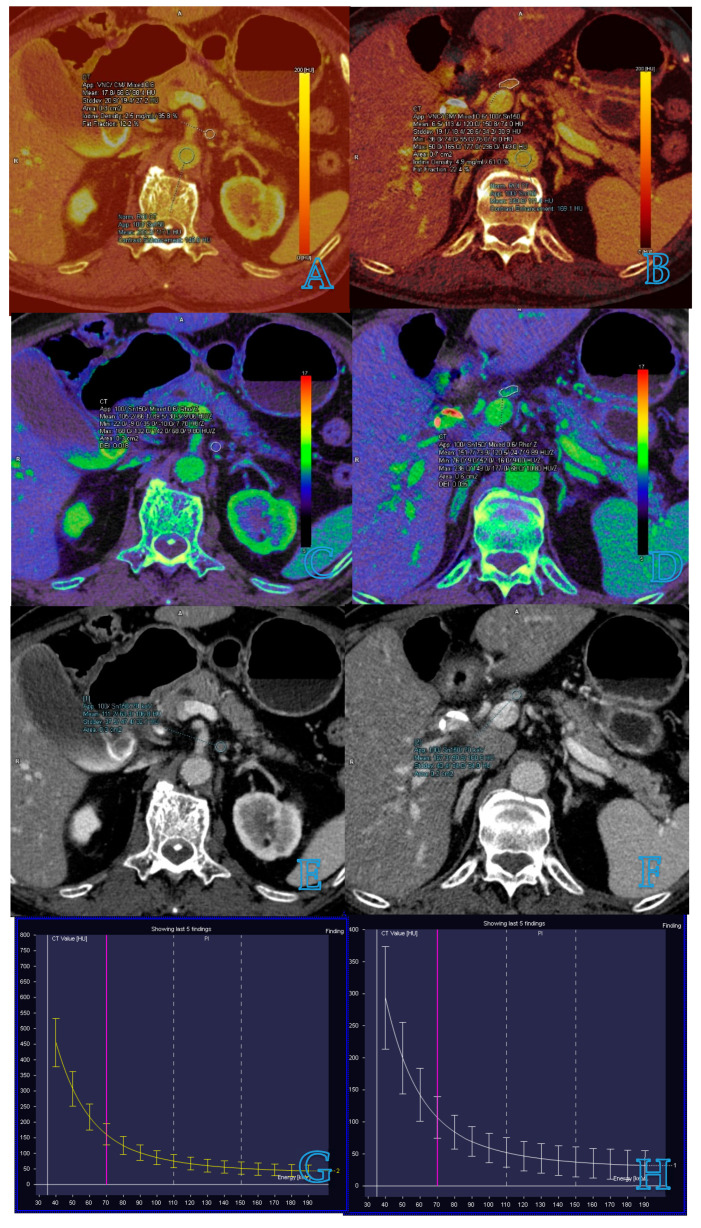
Measurement of dual-energy CT parameters for small lymph nodes in the same patient with pancreatic cancer. Images of benign lymph nodes are in the left column and images of malignant lymph nodes are in the right column. (**A**–**D**) Iodine concentration images and effective atomic number images, which outline the target lymph node and measure iodine concentration and effective atomic number. Venous phase contrast-enhanced dual-energy CT images show the target lymph node and λHU in non-metastatic and metastatic lymph nodes (**E**–**H**).

**Table 1 diagnostics-14-00377-t001:** Quantitative and qualitative parameters derived from DECT.

Parameter	Explanation
iodine concentration (IC)	quantitative parameters, reflect the iodine content of tissues and indirectly reflect blood supply
normalized iodine concentration (NIC)	quantitative parameters, NIC = IC_lymph_/IC_vessel_; avoids the effect of individual differences compared with IC [10].
slope of the spectral Hounsfield unit curve (λHu)	quantitative parameters, λHU = CTvalue_40keV_ − CTvalue_60keV_/60, determined by physical and chemical nature of the substance [11]
electron density (ED)	quantitative parameters, the average number of electrons in a volume unit (typically expressed in e/cm^3^) [12], varies with the location of electrons, elemental composition, and structure of tissue
effective atomic number (Zeff)	quantitative parameters, the interaction cross sections for photoelectric effect and Compton scattering can be approximately expressed as proportional to Z_eff_^n^, where n is between 4 and 5 for photoelectric effect and 1 for Compton [13]
extracellular volume (ECV) fraction	quantitative parameters, quantify the iodine contrast in intravascular and extravascular–extracellular spaces
arterial enhancement fraction (AEF)	quantitative parameters, iodine uptake in arterial phase (AP)/iodine uptake in venous phase (VP) × 100% [14]
attenuation value of virtual monochromatic images (VMI)	qualitative parameters, optimizes both image noise and contrast, allows for monoenergetic contrast attenuation measurement

**Table 2 diagnostics-14-00377-t002:** Post-processing techniques of DECT.

Post-Processing Techniques	Explanation
iodine maps	material decomposition images, display the attenuation characteristics attributable to iodine, serve not only as quantitative indicators of blood supply to tissues, provide insights into the angiogenesis and hemodynamic status of lesions [15]
Z_eff_ map	material decomposition images, a quantitative approach used for calculating Z_eff_, provide not only density but also elemental information of samples [16]
electron density map	material decomposition images, used by TPS softwares for calculating dose distributions, DECT allows better quantification of ED [12]
virtual monochromatic imaging (VMI)	also referred to as “monoenergetic imaging”, generated within the 40–190 keV range, renowned for its ability to optimize image noise and contrast while allowing precise monoenergetic contrast attenuation measurements [17]
spectral Hounsfield unit attenuation curves	serve as a quantitative measure correlating with different energy levels in VMI, represent the energy-dependent changes in attenuation within a region of interest, typically spanning from 40 to 140 keV, varies across different tissues [18]
virtual non-contrast (VNC)	produced by subtracting the iodine map from the dual-energy enhanced CT image, may replace a pre-contrast scan and substantially reduce radiation exposure [8]

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
