# Peer review of "Dual-Energy Computed Tomography in Detecting and Predicting Lymph Node Metastasis in Malignant Tumor Patients: A Comprehensive Review"

_diagnostics, 2024, doi:10.3390/diagnostics14040377_

Round 1
Reviewer 1 Report
Comments and Suggestions for Authors
In this review, the authors give an insight on the current status and development of dua-energy CT in detecting and predicting lymph node metastasis in malignant tumor patients.
They evaluate the clinical relevance and utility of DECT in the diagnosis of lymph node metastases in malignant tumors.
The role of DECT in comparison to conventional imaging modalities including MRI and ultrasound should be analyzed more in detail, comparing specific MR sequences and contrast enhanced ultrasound.
Author Response
Thank you very much for taking the time to review this manuscript. Please find the detailed responses below and the corresponding revisions/corrections highlighted/in track changes in the re-submitted files.
Comments : In this review, the authors give an insight on the current status and development of dua-energy CT in detecting and predicting lymph node metastasis in malignant tumor patients.They evaluate the clinical relevance and utility of DECT in the diagnosis of lymph node metastases in malignant tumors. The role of DECT in comparison to conventional imaging modalities including MRI and ultrasound should be analyzed more in detail, comparing specific MR sequences and contrast enhanced ultrasound.
Response: Thank you very much for your constructive comment. Cited Works in our manuscript focused on US morphological parameters, MR morphological parameters and signal intensity. The role of DECT in comparison to contrast enhanced ultrasound and specific MR sequences needs further research.
Reviewer 2 Report
Comments and Suggestions for Authors
1. The fomular about λ HU in Table 1 is wrong. It should be (CTvalue40keV - CTvalue60keV )/60.
2. Please tell readers which quantitative and qualitative indicators were used respectively.
3. At the end of the first paragraph in section 3.7, it is said that the diagnostic performance of DECT is superior to MRI. However, the examples provided below only demonstrated that there is no significant difference in the accuracy of MRI and DECT in diagnosing rectal cancer staging. There are not any direct evidence to prove that the diagnostic effect of DECT is superior to MRI. What's more, it is discussed that the combination may provide more insights. Please explain more.
Comments on the Quality of English LanguageIt's OK
Author Response
Thank you very much for taking the time to review this manuscript. Please find the detailed responses below and the corresponding revisions/corrections highlighted/in track changes in the re-submitted files.
Comments 1: The fomular about λ HU in Table 1 is wrong. It should be (CTvalue40keV - CTvalue60keV )/60.
Response 1: Thank you very much for pointing out the problem in our manuscript. Correction has been made in the revised manuscript (Page 3, Line 10).
Comments 2: Please tell readers which quantitative and qualitative indicators were used respectively.
Response 2: Thank you very much for your constructive comment. We now add the footnote in Table 1.
Comments 3: At the end of the first paragraph in section 3.7, it is said that the diagnostic performance of DECT is superior to MRI. However, the examples provided below only demonstrated that there is no significant difference in the accuracy of MRI and DECT in diagnosing rectal cancer staging. There are not any direct evidence to prove that the diagnostic effect of DECT is superior to MRI.
Response 3: Thank you very much for pointing out the key problem in our manuscript. Correction has been made in the revised manuscript (Page 7, Line 23).
Comments 4: It is discussed that the combination may provide more insights. Please explain more.
RESPONSE 4: Follow your suggestion, we now explain more detail about the combination of MRI and DECT in the revised manuscript (Page 8, Line 1-4).
Reviewer 3 Report
Comments and Suggestions for Authors
This article provides an overview of recent literature on the clinical use of Dual Energy Computed Tomography, in staging lymph node metastasis in oncological patients. The paper is well written and covers almost all aspects relevant for current practice for identification and staging of metastatic LNs. However, besides some minor issues related to nomenclature, the authors seem to have totally neglected the recent quickly growing spread of photon counting CT (PCCT) in clinical context, as a new tool for obtaining DECT image and which is also being used for LN characterization in the latest years.
More specifically, I shall recommend to make some amendmement to the manuscript before publication:
1) Page 2, Section 2, lines 58-61. Other than the five mentioned methods to implement DECT (and then, iodine maps, VMI, Zeff/rho_e, etc), the authors must also cite PCCT. Differently than other DECT implementation, PCCT uses special X-ray detectors capable of collecting incoming X-ray photons in (>=2) energy bins, without any need for kVp switching of split filters. Several review articles have been published in the last 2-3 years on this subject. They can pick, for instance,
- First Clinical Photon-counting Detector CT System: Technical Evaluation
Kishore Rajendran, Martin Petersilka, André Henning, Elisabeth R. Shanblatt, Bernhard Schmidt, Thomas G. Flohr, Andrea Ferrero, Francis Baffour, Felix E. Diehn, Lifeng Yu, Prabhakar Rajiah, Joel G. Fletcher, Shuai Leng, and Cynthia H. McCollough
Radiology 2022 303:1, 130-138
Notably PCCT-based iodine maps have been used recently (2023) for characterization of LNs in breast cancer, with comparable performance to MRI:
- Mariana Yalon, Tiffany Sae-Kho, Akriti Khanna, Shaojie Chang, Boleyn R Andrist, Nikkole M Weber, Safa Hoodeshenas, Andrea Ferrero, Katrina N Glazebrook, Cynthia H McCollough, Francis I Baffour, Staging of breast cancer in the breast and regional lymph nodes using contrast-enhanced photon-counting detector CT: accuracy and potential impact on patient management, British Journal of Radiology, 2023;, tqad042, https://doi.org/10.1093/bjr/tqad042
which is worth to be cited in this review.
2) Page 5, section 3.4, lines 174-175. Definition of effective Z. Indeed, the definition used by authors is not clear. When dealing with radiation-matter interaction processes, the definition of Z_eff for compounds or mixtures (and with heterogeneous materials in general) involves the creation of a fictitious element with atomic number Z_eff, such that the interaction cross sections for photoelectric effect and Compton scattering can be approximately expressed as proportional to Z_eff^(n), where n is between 4-5 for photoelectric effect and 1 for Compton. Cite, for instance:
- Murty, R. Effective Atomic Numbers of Heterogeneous Materials. Nature 207, 398–399 (1965). https://doi.org/10.1038/207398a0
3) Page 5, section 3.5, lines 186-187. Also in this case, definition of ED is not correct. The provided definition is a quantum mechanical one, suitable for describing electron orbitals based on particle's wave functions. Nothing to do with the quantity used in CT imaging, which is simply the (average) number of electrons in a volume unit (typically expressed in e-/cm^3). ED is especially important in radiation therapy treatment planning, where accurate estimations of CT-derived ED maps are used by TPS softwares to calculate dose distributions. Even though ED can be 1-to-1 mapped to HU in single energy CT, DECT allows better quantification of ED. Cite, for instance.
- Mei, K., Ehn, S., Oechsner, M. et al. Dual-layer spectral computed tomography: measuring relative electron density. Eur Radiol Exp 2, 20 (2018). https://doi.org/10.1186/s41747-018-0051-8
4) Page 5, line 212. "[...]. 80 metastatic [...]". Don't begin sentences with numbers, use "Eighty" instead.
5) Page 8, line 346: "heterogeneous". Did you mean "heterogeneity"?
Author Response
Thank you very much for taking the time to review this manuscript. Please find the detailed responses below and the corresponding revisions/corrections highlighted/in track changes in the re-submitted files.
Comments 1: Page 2, Section 2, lines 58-61. Other than the five mentioned methods to implement DECT (and then, iodine maps, VMI, Zeff/rho_e, etc), the authors must also cite PCCT. Differently than other DECT implementation, PCCT uses special X-ray detectors capable of collecting incoming X-ray photons in (>=2) energy bins, without any need for kVp switching of split filters. Several review articles have been published in the last 2-3 years on this subject. They can pick, for instance,
- First Clinical Photon-counting Detector CT System: Technical Evaluation
Kishore Rajendran, Martin Petersilka, André Henning, Elisabeth R. Shanblatt, Bernhard Schmidt, Thomas G. Flohr, Andrea Ferrero, Francis Baffour, Felix E. Diehn, Lifeng Yu, Prabhakar Rajiah, Joel G. Fletcher, Shuai Leng, and Cynthia H. McCollough
Radiology 2022 303:1, 130-138
Notably PCCT-based iodine maps have been used recently (2023) for characterization of LNs in breast cancer, with comparable performance to MRI:
- Mariana Yalon, Tiffany Sae-Kho, Akriti Khanna, Shaojie Chang, Boleyn R Andrist, Nikkole M Weber, Safa Hoodeshenas, Andrea Ferrero, Katrina N Glazebrook, Cynthia H McCollough, Francis I Baffour, Staging of breast cancer in the breast and regional lymph nodes using contrast-enhanced photon-counting detector CT: accuracy and potential impact on patient management, British Journal of Radiology, 2023;, tqad042, https://doi.org/10.1093/bjr/tqad042
which is worth to be cited in this review.
Response 1: Thank you very much for your constructive suggestions and very useful comments. We now add PCCT and explain more detail about it in the revised manuscript (Page 8, Line 7-13).
Comments 2: Page 5, section 3.4, lines 174-175. Definition of effective Z. Indeed, the definition used by authors is not clear. When dealing with radiation-matter interaction processes, the definition of Z_eff for compounds or mixtures (and with heterogeneous materials in general) involves the creation of a fictitious element with atomic number Z_eff, such that the interaction cross sections for photoelectric effect and Compton scattering can be approximately expressed as proportional to Z_eff^(n), where n is between 4-5 for photoelectric effect and 1 for Compton. Cite, for instance:
- Murty, R. Effective Atomic Numbers of Heterogeneous Materials. Nature 207, 398–399 (1965). https://doi.org/10.1038/207398a0
Response 2: Thank you very much for pointing out the key problem in our manuscript. Correction has been made in the revised manuscript (Page 6, Line 6-11).
Comments 3: Page 5, section 3.5, lines 186-187. Also in this case, definition of ED is not correct. The provided definition is a quantum mechanical one, suitable for describing electron orbitals based on particle's wave functions. Nothing to do with the quantity used in CT imaging, which is simply the (average) number of electrons in a volume unit (typically expressed in e-/cm^3). ED is especially important in radiation therapy treatment planning, where accurate estimations of CT-derived ED maps are used by TPS softwares to calculate dose distributions. Even though ED can be 1-to-1 mapped to HU in single energy CT, DECT allows better quantification of ED. Cite, for instance.
- Mei, K., Ehn, S., Oechsner, M. et al. Dual-layer spectral computed tomography: measuring relative electron density. Eur Radiol Exp 2, 20 (2018). https://doi.org/10.1186/s41747-018-0051-8
Response 3: Thank you very much for pointing out the key problem in our manuscript. Correction has been made in the revised manuscript (Page 6, Line 21-25).
Comments 4: Page 5, line 212. "[...]. 80 metastatic [...]". Don't begin sentences with numbers, use "Eighty" instead.
Response 4: Thank you very much for pointing out the problem in our manuscript. Correction has been made in the revised manuscript (Page 7, Line 4).
Comments 5: Page 8, line 346: "heterogeneous". Did you mean "heterogeneity"?
Response 5: Thank you very much for pointing out the problem in our manuscript. Correction has been made in the revised manuscript (Page 10, Line 13).
Round 2
Reviewer 3 Report
Comments and Suggestions for Authors
Just correct the text at line 186 "Z_eff^(n)", using n as exponent. It should look like Zeffn (better if with 'n' closer to the Z). I used this LaTeX-type notation because I only had a plain text editor during my first review.
Author Response
Thank you very much for taking the time to review this manuscript. Please find the detailed responses below and the corresponding revisions/corrections highlighted/in track changes in the re-submitted files.
Comments : Just correct the text at line 186 "Z_eff^(n)", using n as exponent. It should look like Zeffn (better if with 'n' closer to the Z).
Response : Thank you very much for pointing out the problem in our manuscript. Correction has been made in the revised manuscript (Page 6, Line 9).